# Analysis of *Lin28B* Promoter Activity and Screening of Related Transcription Factors in Dolang Sheep

**DOI:** 10.3390/genes14051049

**Published:** 2023-05-07

**Authors:** Zhiyuan Sui, Yongjie Zhang, Zhishuai Zhang, Chenguang Wang, Xiaojun Li, Feng Xing, Mingxing Chu

**Affiliations:** 1Key Laboratory of Tarim Animal Husbandry Science and Technology, Xinjiang Production and Construction Group, Alar 843300, China; zhiyuan717@126.com (Z.S.); yongjie915@163.com (Y.Z.); 15100373890@163.com (Z.Z.); wcg0330ljn@163.com (C.W.); lxj17699252715@126.com (X.L.); 2College of Animal Science and Technology, Tarim University, Alar 843300, China; 3Key Laboratory of Animal Genetics, Breeding and Reproduction of Ministry of Agriculture and Rural Affairs, Institute of Animal Science, Chinese Academy of Agricultural Sciences, Beijing 100193, China

**Keywords:** Dolang sheep, *Lin28B*, promoter, transcription factor

## Abstract

The *Lin28B* gene is involved in the initiation of puberty, but its regulatory mechanisms remain unclear. Therefore, in this study, we aimed to study the regulatory mechanism of the *Lin28B* promoter by cloning the *Lin28B* proximal promoter for bioinformatic analysis. Next, a series of deletion vectors were constructed based on the bioinformatic analysis results for dual-fluorescein activity detection. The transcriptional regulation mechanism of the *Lin28B* promoter region was analyzed by detecting mutations in transcription factor-binding sites and overexpression of transcription factors. The dual-luciferase assay showed that the *Lin28B* promoter region −837 to −338 bp had the highest transcriptional activity, and the transcriptional activity of the *Lin28B* transcriptional regulatory region decreased significantly after *Egr1* and *SP1* mutations. Overexpression of the Egr1 transcription factor significantly enhanced the transcription of *Lin28B*, and the results indicated that *Egr1* and *SP1* play important roles in regulating *Lin28B*. These results provide a theoretical basis for further research on the transcriptional regulation of sheep *Lin28B* during puberty initiation.

## 1. Introduction

Puberty in female animals is the period when estrus appears for the first time and ovulation occurs. It plays an important role in animal growth and development. In addition, this is an important phase for animals to acquire their reproductive ability [1]. Puberty onset is genetically and nongenetically controlled by interactions between factors [2]. In the early stages, the hypothalamic tissue of Dolang sheep was used to screen the gene *Lin28B* in regard to the onset of puberty through transcriptome sequencing technology and high expression was detected in the hypothalamus, pituitary, ovaries, and fallopian tubes through qPCR and Western blot technology. The *Lin28B* gene has a certain relationship with the initiation of puberty in Dolang sheep [3].

The *Lin28B* gene was first cloned from human hepatocellular carcinoma cells. It is overexpressed in hepatocellular carcinoma and is highly similar to the heterochronous gene *Lin28.* These two genes may originate from the duplication of the same ancestral gene. Their mRNAs have a long 3′-UTR with the complementary site of *Let7* microRNA, which indicates that these genes may be the natural target of *Let7*. *Lin28B* contains a cold shock domain (CSD) and a pair of CCHC zinc finger domains [4]. Related studies have shown that *Lin28B* is involved in the regulation of puberty [5,6]. Sangiao-Alvarellos et al. (2013) found that *Lin28B* mRNA was abundantly expressed in the hypothalamus of neonatal male and female rats and that *Lin28B* expression decreased sharply during the transition from infancy to puberty [7]. Grieco et al. (2013) found that *Lin28B* expression decreases in the ovaries of female mice from infancy to puberty [8].

The promoter is usually located upstream of the 5′end of the structural gene, and is a nucleotide sequence that RNA polymerase recognizes and specifically binds to in the template DNA to ensure the accuracy of transcription initiation [9]. Studies have shown that gene regulation at the transcriptional level is through the combination of TFBS DNA sequences, the sites of the interaction between promoters and TFs [10]. Transcription factors are proteins that regulate gene expression by binding to specific sequences on DNA to activate or repress gene transcription. The regulatory mechanism of transcription factors includes two aspects: one is the regulation of the expression and activity of transcription factors, and the other is the binding selectivity of transcription factors and DNA sequences. In cells, the expression and activity of transcription factors are affected by many factors, such as the activation of cell signaling pathways and protein modification. The binding selectivity of transcription factors and DNA sequences is also determined by many factors, such as the structure of the transcription factors, the specificity of DNA sequences, etc. The regulatory mechanisms of transcription factors are critical for the normal function and development of cells [11].

At present, there is no related research on the promoter region of sheep *Lin28B* in the literature. In order to clarify the significance of the promoter region to the transcriptional regulation of the *Lin28B*, this study constructed a promoter deletion fragment vector to explore the core promoter region. To further elucidate the transcription regulation mechanism of *Lin28B,* this study predicted and verified the related transcription factors *SP1* and *Egr1*, thus providing a theoretical basis for the regulation mechanism of sheep *Lin28B* in puberty initiation.

## 2. Materials and Methods

### 2.1. Ethics Statement

All animal experiments were approved by the Animal Protection and Use Committee of the Tarim University of Science and Technology College (Approval No. 2020.050).

### 2.2. Sample Collection

Under natural light, all Dolang sheep were given free access to food and water and were healthy. Fifteen three-month-old female Dolang sheep maintained at the Tarim University experimental station were used as models in this study. Blood (5 mL) was collected from the jugular vein of each sheep, treated with EDTA-K anticoagulant (5:1 ratio of blood to EDTA-K (*V*/*V*)), and stored in a refrigerator at −20 °C for later use. Genomic DNA was extracted from Dolang sheep blood using a DNA extraction kit (Tiangen, Beijing, China) according to the manufacturer’s instructions. Its purity was detected using 1.5% agarose gel electrophoresis, and its content was detected using an ultraviolet spectrophotometer. The extracted DNA was stored at −20 °C.

### 2.3. Lin28B Promoter Sequence Analysis and Vector Construction

Using the promoter sequence of the sheep *Lin28B* gene in the Ensembl database as a template, the Primer Premier 6.0 software (Premier, North Carolina, Canada) was used to design primers to amplify 7 promoter fragments of different lengths, namely L1 (−2837 to +162), L2 (−2337 to +162), L3 (−1837 to +162), L4 (−1337 to +162), L5 (−837 to +162), L6 (−337 to +162), and L7 (−837 to −338). The restriction endonuclease sites *Nhe*I and *Xho*I (Takara, Beijing, China) were added to the upstream and downstream primers, respectively, and protective bases were added before the enzyme cutting sites. Primer sequences are listed in Table 1. PCR reaction (25 μL): DNA 1.0 µL, upstream and downstream primers (10 µmol/L) each 1.0 µL, 2× EasyTaq PCR SuperMix (+dye) (TransGen, Beijing, China) 12.5 µL, sterile deionized water (ddH_2_O) 9.5 µL. PCR conditions: pre-denaturation at 94 °C for 5 min; denaturation at 94 °C for 30 s, annealing at 60 °C for 30 s, extension at 72 °C for 2 min, 40 cycles; extension at 72 °C for 10 min.

### 2.4. Vector Construction and Transfection

The PCR product was subjected to electrophoresis gel, and the target fragment was recovered using an agarose gel recovery kit (Tiangen, Beijing, China). The recovered product was ligated to the *pGL3-Basic* vector; ligation system (10 µL): 4 µL of recovered PCR product, 1 µL of vector, 5 µL of solution I. The ligation reaction was carried out at 4 °C for over 12 h. A volume of 5 µL of the ligated product was added to 100 µL DH5α cells (TransGen, Beijing, China); after heat-shock in a 42 °C water bath for 90 s, the centrifuge tube was quickly transferred to an ice bath for 1–2 min. Liquid LB medium (395 µL) was added, and the mix was shaken at 200 rpm at 37 °C for 90 min. Then, 200 µL of LB medium containing DH5α was added to AMP medium at 37 °C and cultured for 12–16 h. Single colonies were selected for culture and plasmid purification. Colony screening was performed by polymerase chain reaction (PCR) and double enzyme digestion (*Nhe*I/*Xho*I) experiments. The positive plasmid was sent to the Shanghai Sangon Bioengineering Company for sequencing. Plasmids with the correct sequence were amplified in large quantities and stored at −20 °C for later use.

### 2.5. Cell Transfection and Dual-Luciferase Detection of Deletion Fragment Luciferase Reporter Gene Vector

Cultured 293T cells were suspended in PBS for later use. Trypan blue staining solution (0.4%) was added to the cell suspension at a cell dye ratio of 1:1 (*v*/*v*), and the cells were counted using a hemocytometer. The day before transfection, a total of 1 × 10^4^ 293T cells (provided by the Chinese Academy of Sciences Cell Bank) were inoculated in 96-well plates, and 150 µL freshly prepared complete medium (89% DMEM+ 10% fetal bovine serum) (Gibco, California, USA) was added, mixed, and gently shaken to prevent cell accumulation in the center of the well that could result in mass cell death. They were cultured at 37 °C with 5% CO_2_ for subsequent experiments. When the confluence of the cells reached 70–80%, the cells were co-transfected with the target plasmid and the internal reference plasmid *pRL-TK* in a ratio of 10:1 according to the instructions of the Lipofectamine™ 3000 transfection reagent (Invitrogen, California, USA). At least three biological replicates were performed. The 96-well plate was taken out of the incubator 36 h after transfection, and the dual-luciferase activity was detected according to the instructions of the Dual-Glo^®^ Luciferase Assay System kit (Promega, Madison, USA). The activity of firefly luciferase and Renilla luciferase were detected with a Synergy H1 chemiluminescence detector (Biotek, Vermont, USA).

### 2.6. Transcription Factor Prediction of Highly Transcriptionally Active Fragments

Using a combination of the online software JASPAR 2022 [12] and Animal TFDB 3.0 (Huazhong University of Science and Technology, Wuhan, China) to predict the L7 (−837 bp to −338 bp) fragment of the *Lin28B* promoter, the potential binding sites of transcription factors related to the regulation of puberty with higher scores were obtained.

### 2.7. Construction of Point Mutation Vectors for Transcription Factors SP1 and Egr1

The assay was performed according to the manufacturer’s instructions for the Express Point Mutation Kit (Tiangen, Beijing, China). The primer design instructions from the Primer Premier software (version 6.0) were used to design the transcription factor *Egr1* primers ETF1 and ETR1, transcription factor *SP1* primers STF2 and STR2, and transcription factor point mutation primers (Table 2). Using the L7 fragment as a template, *Egr1*-binding-site-mutant plasmid ET1 and *SP1*-binding-site-mutant plasmid ST2 were constructed.

According to the instructions of the Lipofectamine™ 3000 transfection reagent, 293T cells were co-transfected with *pRL -TK* (Promega, Madison, USA) and *pGL3-basic*, plasmid L7, mutant plasmids ET1 or ST2; at least three biological replicates were set up for each treatment. The 96-well plate was removed from the incubator 36 h after transfection, and the dual-luciferase activity was detected according to the instructions of the Dual-Glo Luciferase Assay System kit.

### 2.8. Construction of Transcription Factor Egr1 Overexpression Vector and Dual Luciferase Detection

The *Egr1* overexpression vector was constructed by Suzhou GENEWIZ Co., Ltd. (Suzhou, Jiangsu, China). The *Egr1* overexpression vector, *pEGFP-N1*, and plasmid L7 were co-transfected into 293T cells according to the instructions of the Lipofectamine™ 3000 transfection reagent, and at least three biological replicates were set up for each treatment. The 96-well plate was removed from the incubator 36 h after transfection, and the dual-luciferase activity was detected according to the instructions of the Dual-Glo Luciferase Assay System kit.

### 2.9. Data Analysis

Relative activity value of dual luciferases = firefly luciferase activity (F value)/Renilla luciferase activity values (R value). The data were analyzed using SPSS (IBM, Almon, New York, USA) and expressed as the standard error of the mean. At least three biological replicates were set up for each treatment group, and the statistical analysis was performed using a one-way ANOVA. * *p* < 0.05 and ** *p* < 0.01 indicated significant and extremely significant differences, respectively.

## 3. Results

### 3.1. Construction of the Deletion Vector of Lin28B Gene Promoter Fragment in Dolang Sheep

As shown in Figure 1, using the sheep *Lin28B* promoter fragment as a template, a series of deletion primers was used to clone Lin 28 gene promoter deletion fragments of different lengths: L1 (−2837 to +162), L2 (−2337 to +162), L3 (−1837 to +162), L4 (−1337 to +162), L5 (−837 to +162), L6 (−337 to +162), and L7 (−837 to−338). The missing fragment of the *Lin28B* promoter was ligated to a luciferase reporter gene vector to obtain the corresponding recombinant vector.

### 3.2. Detection of Dual-Luciferase Activity in the Deletion Vector of Lin28B Gene Promoter Fragments in Dolang Sheep

After incubation for 36 h, the promoter activity of each fragment was detected (Figure 2). The results showed that the L1 and L6 promoters had the lowest activity, whereas the L2 and L5 promoters had the highest activity. Furthermore, the results found that with the deletion of −887 bp to −338 bp, the promoter activity dropped suddenly and decreased to a half. The above results indicate that the −887 to −338 bp region may play a critical role in maintaining the transcriptional activity of *Lin28B*.

### 3.3. Dual Luciferase Activity Detection of Fragment Vectors with High Transcriptional Activity

The results of the dual luciferase assay (Figure 3) showed that the promoter activity of L5 was approximately the same as the sum of the promoter activities of L6 and L7, and the promoter activities of L5 and L7 were significantly higher than that of L6. The above test results showed that the L7 (−887 to −338) fragment is the core promoter region.

### 3.4. Prediction of Transcription Factor Binding Sites in Lin28B Promoter Region of Dolang Sheep

The online software JASPAR and Animal TFDB 3.0 were used to predict the transcription factor binding sites in the *Lin28B* promoter L7 (−837 bp to −338 bp) fragment and the transcription factors related to the transcriptional regulation of puberty with higher scores were selected. The results are summarized in Table 3. Two transcription factors were predicted, namely transcription factors *SP1* and *Egr1*.

### 3.5. Verification of the Point Mutation of the Transcription Factor Binding Sites in the Core Region of the Lin28B Promoter of Dolang Sheep

The mutant vectors were transfected into 293T cells, and the L7 fragment was used as a control. The promoter activities of the mutant and control promoter groups were detected using a double luciferase assay, and the differences were compared. The results showed that the activity of the L7 fragment with *Egr1* and *SP1* mutations compared with the non-mutated fragment was extremely significantly different (*p* < 0.01), and the difference between the L7 fragment and the EGRI mutant vector was more significant (Figure 4).

### 3.6. Effect of Egr1 Overexpression on the Transcriptional Activity of Lin28B Promoter

The results showed that the fluorescence activity ratio of the L7 fragment co-transfected with the *Egr1* overexpression vector was significantly higher (*p* < 0.01) than that of the control group co-transfected with the *pEGFP-N1* and L7 fragments (Figure 5), indicating that overexpression of *Egr1* has an impact on the transcriptional activity of the L7 fragment.

## 4. Discussion

Puberty is the period when the animal’s reproductive ability begins, and the timing of puberty is also related to the animal’s reproductive ability. Animals with early puberty have a higher reproductive ability and more offspring throughout their life. Studies have confirmed that there is a certain correlation between the expression of *Lin28B* mRNA and the puberty of domestic animals [1]. However, there is no specific report on the regulatory mechanism of the *Lin28B* gene in puberty in sheep. The identification of promoters is key to the regulation of gene transcription [13]. Luciferase analysis is a common method for studying promoter core and regulatory regions [14,15]. The dual-luciferase reporter assay uses the firefly luciferase (*pGL3*) to kidney luciferase (*pRL-TK*) ratio to determine gene expression. This method can reduce experimental errors caused by factors such as cell activity, transcription efficiency, and cleavage efficiency, and the test results are reliable and accurate [16]. In this study, a promoter deletion fragment vector was constructed and transfected into cells. Double luciferase detection revealed that the L1 (−2837 to +162) and L6 (−337 to +162) vectors had the lowest activity, whereas the L2 (−2337 to +162) and L5 (−837 to +162) were the most active promoters. There was a significant difference in the promoter activity of L7 (−837 to −338 bp) and L5 (−837 to +162 bp), and the core promoter region was generally located in the upstream 500 bp [17]. According to previous studies, there is a CpG island in this region, which is related to the expression of the *Lin28B* gene and the mRNA expression in the process of puberty initiation in Dolang sheep. Thus, we inferred that L7 (−837 to −338 bp) is the core promoter region of the gene.

A gene promoter is a complex region containing many transcription factor binding sites which determine the transcription start site and frequency [18]. Transcription factors (TFs) can specifically recognize the active region of the gene promoter, thereby activating or inhibiting the transcriptional activity of the target gene, regulating the transcription levels of the gene, and affecting the protein’s function [19]. Transcriptional regulation of genes is affected by the activity of different transcription sites in the promoter region. In this study, JASPAR and Animal TFDB 3.0 were used to predict the transcription factor binding sites in the *Lin28B* promoter L7 (−837 bp to −338 bp) fragment, and the intersection of higher scores was selected to obtain the two transcription factors *Egr1* and *SP1*. The *Erg1* transcription factor was point-mutated in this study, and an overexpression vector was constructed. Double-luciferase detection results showed that the *Egr1* transcription factor plays an important role in the transcriptional regulation of *Lin28B.* According to relevant studies, *Egr1* plays an important role in the reproductive system, and *Egr1* can maintain sufficient LHb expression in the female pituitary [20,21]. During proestrus in rats, GnRH stimulates the expression of *Egr1* and simultaneously binds *Egr1* to two conserved cis-acting elements of the LH proximal promoter [22,23,24]. *Egr1* knockout mice showed reduced fertility, blocked GnRH-induced pituitary LH production, and impaired ovarian LH receptor expression. *Egr1* regulates the expression of many important factors involved in folliculogenesis and ovulation in the ovary [20,25,26]. Previous studies have shown that *Egr1* is involved in various cell biological processes, such as differentiation, proliferation, and apoptosis, and *Lin28B* is a key gene involved in the development of puberty in sheep. Through research, it was found that *Egr1* can combine with the promoter region of *Lin28B* gene and significantly increase the promoter activity of *Lin28B* gene. Therefore, this study suggests that the *Egr1* transcription factor plays an important role in the transcriptional regulation of *Lin28B* during puberty in sheep. In this study, we constructed a point mutation vector for the transcription factor *SP1.* After testing, it was found that the *SP1* transcription factor was found to affect the transcriptional regulation of *Lin28B.* Studies have reported that *SP1* interacts with various epigenetic factors, actively regulates gene transcription, and promotes gene expression [27]. Further studies have shown that *SP1* interacts with epigenetic factors that negatively regulate gene transcription to inhibit gene expression [28,29]. During primordial follicle formation in the ovary, *SP1* is present in oocytes and somatic cells. Knockdown of *SP1* expression, especially in pregranulosa cells, significantly inhibits oocyte apoptosis and primordial follicle formation, suggesting that somatically expressed *SP1* plays a role in primordial folliculogenesis, clarifying that *SP1* regulates pregranulosa cells during mammalian ovarian development to control the establishment of ovarian reserves [30]. At present, most of the mechanisms for the initiation of puberty are estrogen feedback mechanisms. When approaching puberty, gonadotropin-releasing hormone (GnRH) can activate the established gonadal axis [31], the pulsatile secretion of GnRH is enhanced, the secreted GnRH is retained in the blood, and enters the adenohypophysis through the pituitary portal system, thereby promoting luteinizing hormone (LH) and follicle-stimulating hormone (FSH) secretion [32]. This causes the levels of LH and FSH in the blood to start rising. At this time, LH and FSH in the blood work together to stimulate the growth and development of follicles; LH can also stimulate theca cells to secrete a large amount of androgens, and the aromatase produced by FSH stimulates granulosa cells to ingest and synthesize estradiol (E2) [33], thereby inducing the central nervous system to secrete GnRH and LH and promote ovulation to complete the initiation process of puberty. Therefore, our results suggest that *SP1* plays a role in the transcriptional regulation of *Lin28B* during puberty in sheep.

## 5. Conclusions

In this study, by constructing promoter fragment deletion vectors, we found that −887 to −338 bp is a region with high transcriptional activity. To further understand the transcriptional regulatory mechanism of *Lin28B*, we constructed a transcription factor point mutation vector and an *Egr1* overexpression vector. Double-luciferase assays revealed that the transcription factors *SP1* and *Egr1* can significantly affect the activity of the promoter. Therefore, this study suggests that the transcription factors *SP1* and *Egr1* play a role in the transcriptional regulation of *Lin28B*.

## Figures and Tables

**Figure 1 genes-14-01049-f001:**
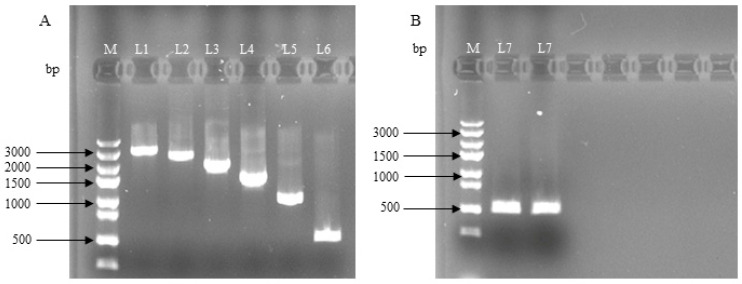
Dolang sheep *Lin28B* promoter fragment deletion vector electrophoresis map. M, marker 5000; **A**: Electropherogram of PCR product of promoter deletion fragment (L1~L6); **B**: Electropherogram of promoter fragment (L7) PCR product; L1 (−2837 to +162), L2 (−2337 to +162), L3 (−1837 to +162), L4 (−1337 to +162), L5 (−837 to +162), L6 (−337 to +162), and L7 (−837 to −338).

**Figure 2 genes-14-01049-f002:**
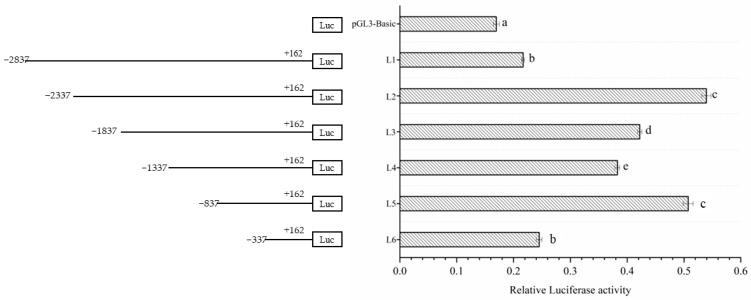
The activity analysis of different Dolang sheep *Lin28B* promoter lengths. The left side is a schematic diagram of the length of each deletion fragment, and the right side is the promoter activity of the corresponding fragment; different lowercase letters represent highly significant differences (*p* < 0.01).

**Figure 3 genes-14-01049-f003:**
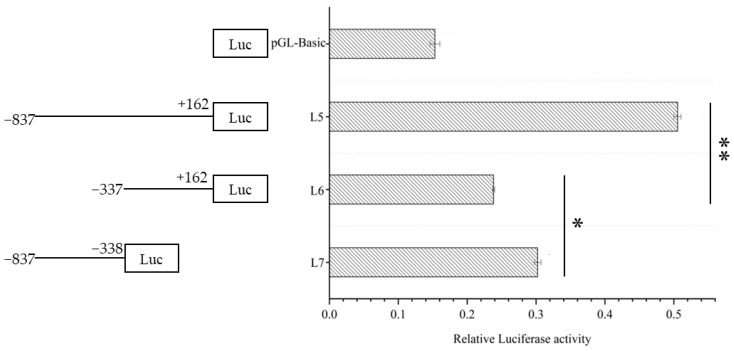
The activity analysis of different lengths of Dolang sheep *Lin28B* promoter. The left side is a schematic diagram of the length of each deletion fragment, and the right side is the promoter activity of the corresponding fragment. The marker “**” represents a highly significant difference (*p* < 0.01). The marker “*” represents a significant difference (*p* < 0.05).

**Figure 4 genes-14-01049-f004:**
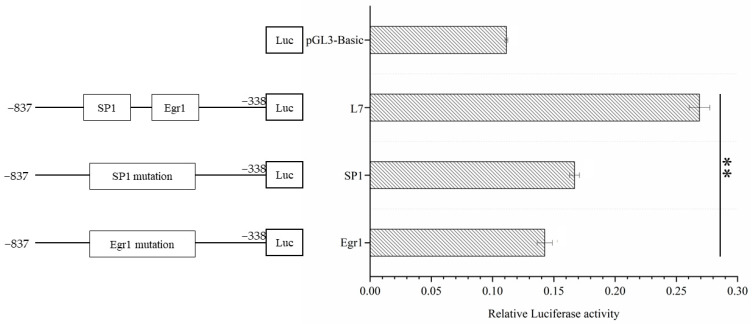
The point mutation analysis of *Egr1* and *SP1* binding sites in the L7 fragment. *SP1*: the *SP1* mutation. *Egr1*: the *Egr1* mutation. The marker “**” represents a highly significant difference (*p* < 0.01).

**Figure 5 genes-14-01049-f005:**
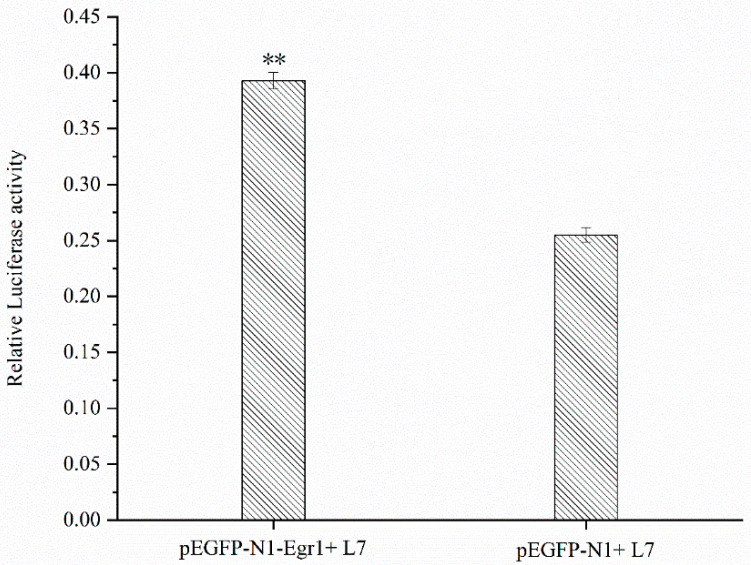
Promoter activity analysis of *Egr1* overexpression. For *pEGFP-Egr1* and L7, the *Egr1* overexpression vector was co-transfected with the L7 fragment; for *pEGFP-N1* and L7, the pEGFP-N1 vector was co-transfected with the L7 fragment. The group of *pEGFP-N1* and L7 was the control group. The marker “**” represents a highly significant difference (*p* < 0.01).

**Table 1 genes-14-01049-t001:** Primers used for fragment deletion of sheep *Lin28B* gene promoter constructs.

Primer Name	Primer Sequence (5′–3′)	Product Length (bp)
L1	F: *CTAgctagc*GATAACCAACGGGCATTTA R: *CCGctcgag*CTGGCAAGAGGAAGAGATAAC	3012
L2	F: *CTAgctagc*CTCCCTCTCCTCTCCCCTCCT CTCCTTTCTTTTACTCTCCACA R: *CCGctcgag*CTGGCAAGAGGAAGAGATAAC	2512
L3	F: *CTAgctagc*TTTATGATAATTTTACATAATGA CATGTCCTCA R: *CCGctcgag*CTGGCAAGAGGAAGAGATAAC	2012
L4	F: *CTAgctagc*TATACATCCATTTATTTC AGATCTGAACTAATTAATTGTCCAT R: *CCGctcgag*CTGGCAAGAGGAAGAGATAAC	1512
L5	F: *CTAgctagc*AGGGACGGTAGGAGCCT AATCCGTTATT R: *CCGctcgag*CTGGCAAGAGGAAGAGATAAC	1012
L6	F: *CTAgctagc*CAGGGCACAATCAGGTACT TGTGT R: *CCGctcgag*CTGGCAAGAGGAAGAGATAAC	512
L7	F: *CTAgctagc*AGGGACGGTAGGAGCCTAAT CCGTTATT R: *CCGctcgag*AAATATTTCTCAAATTTAAAAT AAAATCCTACCGGAAAAATCGCT	512

Capital letters in italics, protecting bases; small italic letters, enzyme recognition sites.

**Table 2 genes-14-01049-t002:** Sequences of site-directed mutagenesis primers for Dolang sheep *Lin28B* gene promoter.

Primer Name	Primer Sequence (5′-3′)
ETF1	F: GCCTGCCTGGCGCTCCCTTCTCCCTCCCTGCCCATACATA
ETR1	R: TATGTATGGGCAGGGAGGGAGAAGGGAGCG CCAGGCAGGC
STF2	F: GGCCCCTCTTTTTCACCTCGCTCCCCAGCCTCGCCCGCTG
STR2	R: CAGCGGGCGAGGCTGGGGAGCGAGGTGAAAAAGAGGGGCC

Note: the underline indicates the mutation site.

**Table 3 genes-14-01049-t003:** Prediction of transcription factor binding sites in the regulatory region of the *Lin28B* promoter in Dolang sheep.

Transcription Factors	Start	Stop	Strand	Matched Sequence (5′-3′)
*SP1*	142	151	+	CCCCTCCCCC
*SP1*	219	228	+	CCCCTCCCCC
*Egr1*	218	231	+	TCCCCTCCCCCTCC

## Data Availability

Not applicable.

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
