# Peer review of "Analysis of Lin28B Promoter Activity and Screening of Related Transcription Factors in Dolang Sheep"

_genes, 2023, doi:10.3390/genes14051049_

Round 1
Reviewer 1 Report
line 136 F and R stands for what????
please add them
when explaining the targeted genomic region please write the paragraph in more understandable way
the procedures used are very good and the conclusion :
in conclusion
when writing "Therefore, this study suggests that the transcription factors SP1 and Egr1 288 play a role in the transcriptional regulation of Lin28B during sheep puberty,"
I hope to see a recommendation to apply those genetic modifications and their true effect on sheep puberty.
when explaining the targeted genomic region please rewrite the paragraph in more understandable way
Reviewer 2 Report
The manuscript number: genes-2365015 entitled ‘Analysis of Lin28B promoter activity and screening of related transcription factors in Dolang sheep’ is aimed to investigate the regulatory mechanism of the Lin28B promoter. The subject of presented paper is interesting, but in reviewer opinion this manuscript should be improved. Below are the comments and questions to the authors that influenced my decision.
Introduction
Point 1. Lines 30, 35 and elsewhere: The Lin28B name should be written in the same way in whole manuscript. Please check the writing of this word in your publication.
Point 2. Lines 48, 50 and elsewhere: Please check carefully the writing of citation. Once you use ‘space’ between citation number and word and in other place you do not do that.
Materials and methods
Point 3. Line 63: Please add more information about animals used in this experiment (age, period time, household conditions etc.).
Point 4. Authors state that they use 2× EasyTaq PCR Su-83 perMix for prepering cDNA library. I could not find any information about this reagent so please tell me those this mix allow to amplify so long product as you state in manuscript and table 1?
Point 5. Line 104: Please add more information about the conditions of cell cultures.
Point 6. Line 116: Why authors use porcine BMP7 promoter to show information on site-directed mutation in primers?
Results
Point 7. All results should be re written as authors in this description to many times repeat the information contained in the methodological description as well as some information which should be used in discussion section. Because of that it is difficult to understand this section.
Discussion
Point 8. The discussion section is rather laconic, and contains methodical repetitions used earlier in the work. I suggests that authors should expanding description of this chapter.
Conclusion
Point 9. In conclusion authors state that ‘the transcription factors SP1 and Egr1 288 play a role in the transcriptional regulation of Lin28B during sheep puberty’ but they never state that they study puberty time in sheep, so this conclusion is inadequate and should be re-written.
Reviewer 3 Report
The paper deals with the Lin28B promoter activity. It is of good quality and brings new knowledge. However, some imperfections must be revised.
Table 1, comment the restriction site sequences in the footnote.
Table 2, porcine BMP7 promoter?
Write the section 2.4 a bit more thoroughly, in detail.
Section 3.2, r. 162, “decreased approximately once”, or “to a half”?
Figure 2, the small letters sign the significancy of differences, i. e. each promoter is significant compared to each other?
Similarly for Figure 3.
Figure 4, legend is missing. You write on the highly significant differences in the text (r. 205, 206, 207), but write small letters in the Figure.
Similarly Figure 5, legend is missing.
Discussion section, rewrite rows 249-258, you repeat just the Results.
Formal errors
Check the writing of “Lin28B” gene in italic throughout the text.
Endonuclease “Nhe” in italic, “I” normal type.
R. 244, lowest activity, not least.
